# Using Resistin, Glucose, Age and BMI and Pruning Fuzzy Neural Network for the Construction of Expert Systems in the Prediction of Breast Cancer

**Vinícius Jonathan Silva Araújo** [1,†] , **Augusto Junio Guimarães** [1,†] ,
**Paulo Vitor de Campos Souza** [1,2,*,†] , **Thiago Silva Rezende** [1] **and Vanessa Souza Araújo** [1]

[1]  Information Systems Course—Centro Universitário UNA de Betim, Minas Gerais 32.510-010, Brazil;
    vinicius.j.s.a22@hotmail.com (V.J.S.A.); augustojunioguimaraes@gmail.com (A.J.G.);
    silvarezendethiago@hotmail.com (T.S.R.); v.souzaaraujo@yahoo.com.br (V.S.A.)
[2]  Federal Center for Technological Education of Minas Gerais, Minas Gerais 30.421-169, Brazil
*  Correspondence: goldenpaul@informatica.esp.ufmg.br
†  These authors contributed equally to this work.

**Abstract:** Research on predictions of breast cancer grows in the scientific community, providing data on studies in patient surveys. Predictive models link areas of medicine and artificial intelligence to collect data and improve disease assessments that affect a large part of the population, such as breast cancer. In this work, we used a hybrid artificial intelligence model based on concepts of neural networks and fuzzy systems to assist in the identification of people with breast cancer through fuzzy rules. The hybrid model can manipulate the data collected in medical examinations and identify patterns between healthy people and people with breast cancer with an acceptable level of accuracy. These intelligent techniques allow the creation of expert systems based on logical rules of the IF/THEN type. To demonstrate the feasibility of applying fuzzy neural networks, binary pattern classification tests were performed where the dimensions of the problem are used for a model, and the answers identify whether or not the patient has cancer. In the tests, experiments were replicated with several characteristics collected in the examinations done by medical specialists. The results of the tests, compared to other models commonly used for this purpose in the literature, confirm that the hybrid model has a tremendous predictive capacity in the prediction of people with breast cancer maintaining acceptable levels of accuracy with good ability to act on false positives and false negatives, assisting the scientific milieu with its forecasts with the significant characteristic of interpretability of breast cancer. In addition to coherent predictions, the fuzzy neural network enables the construction of systems in high level programming languages to build support systems for physicians' actions during the initial stages of treatment of the disease with the fuzzy rules found, allowing the construction of systems that replicate the knowledge of medical specialists, disseminating it to other professionals.

**Keywords:** breast cancer; fuzzy neural network; pruning method; pattern classification

## 1. Introduction

In the human body, there are several types of tissues formed by a plurality of cells. The inharmonious and vertiginous growth of these cells can cause a tumor, being able to be benign or malignant thus originating the cancer [1]. The previous recognition of the disease influences greatly the results of the

treatment, knowing that the delay in the diagnosis can cause severe consequences. Another important point is that cancer patients at the same stage of the disease may have different responses to treatment [2]. With the advancement of technology and science, there are several ways, procedures, and techniques for the detection of cancer pathology. The first examination continues to be the self-test, in which the woman touches her breast by looking carefully at the appearance of abnormal masses. Another widely used technique is screening mammography, which uses X-rays to detect tumors and other breast irregularities more accurately [3]. There is also the biopsy that is based on collecting a small amount of tissue or cells from the suspected area, being analyzed later by a pathologist [4]. A biopsy is an invasive, painful procedure that can be expensive depending on the context. Many studies have been proposed to support this type of approach, including the work of [5] a systemic review was performed on the use of mobile applications to guide and collect patient data on this type of surgical procedure. Finally, another great method is the blood test, allowing several tests to be performed to check the cells of the body, evaluating whether they are with expected values. To assist in this type of treatment that can be highly invasive and harmful to women, much research has been conducted with the aim of collecting databases to assist in the prediction of breast cancer [6]. Researchers around the world are working to improve predictions and consequently treatments for people with breast cancer. The work presented by [7], presents significant attributes of a group of women evaluated as likely to have breast cancer. For the prediction factors, the author used some parameters collected from the samples collected in the physical and biological examination of the participants. After collecting and extracting various characteristics such as glucose and body mass index he applied statistical attribute association and feature selection techniques to determine a viable set of dimensions to aid in the prediction of cancer diagnoses. After collecting the data and choosing the most representative aspects of the problem, the author worked with binary pattern classifications tests, where the values were to find out if the person had the disease or not. To validate their model and their experiments, intelligent tests were performed using regression and linear decision tree models. The technique obtained excellent results and served for other researchers to deepen their research on the dimensions collected. His approach presented values of sensitivity, specificity, and high AUC, but the two proposals initially used in his paper do not generate knowledge due to the black box nature that the intelligent models used to have [7].

Some intelligent models use the data of a problem to extract knowledge about the problem. To unite the concepts of interpretability of fuzzy system concepts, proposed by Lord Zadeh [8] with the main training techniques of the artificial neural networks, hybrid models called fuzzy neural networks were developed [9].

The fuzzy neural networks (FNN) have diverse approaches in the literature, covering the resolution of several complex problems in numerous areas of industry, science, health and others.

The use of fuzzy neural networks has been practiced in several areas, such as electric cars [10], in the area of image recognition [11], to forecast short-term load in the area of electrical engineering [12], selection of characteristics in the categorization of texts [13] and in the health area as in the treatment of HPV through immunotherapy [14]. More recent approaches act on a servo drive system of the synchronous reluctance engine position [15], prediction of wind energy under uncertain data [16], tropical cyclones [17], forecast German economic indices [18], time series forecast [19], and so on.

The proposal of this paper is based on the use of a hybrid structure that integrates concepts of neural networks, fuzzy systems, and pruning methods. The objective of this work is to find a hybrid model with high accuracy in the prediction of breast cancer and at the same time extract knowledge from the database through fuzzy rules, create a specialist system with a high degree of interpretability in the detection of sick people. The basis of information submitted to the model comes from a study proposed by [7]. The architecture of the model used in this paper was initially introduced in [20] and consists of the union of a fuzzy system of inference and a neural network of aggregation. In the first layer, the model uses

fuzzy neurons composed with the centers and sigma values of the division of the input space by the fuzzification technique called Anfis [21], capable of generating equally spaced membership functions. In the second layer, logical neurons are responsible for aggregating the neurons of the first layer. Since the number of neurons may be disproportionate to solve the problem, a pruning approach is used to eliminate neurons less significant to the problem. This technique uses the concepts of f-scores applied in intelligent models by [22]. The training of the fuzzy neural network proposed in [20] is based on extreme learning machine concepts [23] to generate the weights of the neural aggregation network which in turn has a single neuron (can be considered a singleton) can identify the binary outputs of the net. To verify the efficiency of the approach, other models commonly used in the literature will be used to prove the feasibility of the proposal.

The paper is ordered as follows: In the next Section 2 we have the theoretical reference, with a review of the important concepts that will be raised in the course of all work. In Section 3 we discuss the process of using the hybrid artificial intelligence model to improve the predictive results of the expert system, with details of specific processes and concepts used. Section 4 shows the functioning of the tests performed, including database, the experiments used for the evaluations and their respective results. The paper finishes in Section 5 where the final conclusions are presented.

## 2. Theoretical Reference

### 2.1. Breast Cancer

Breast cancer is a disease caused by the disordered spread of cells in this region, thus giving rise to abnormal cells forming the tumor. The primary sign is the appearance of nodules in the region of the six and changes in the physical aspect of the breast. This type of pathology is the most common among women in the United States, accounting for almost one in three cases, accounting for more than 40,000 deaths per year [24].

The late age of first pregnancy, menopause and the early age of menarche are linked to a considerable increase in the development of breast cancer. Multiple pregnancies may modestly lower the risk of acquiring this disease when diagnosed after age 40, regardless of the age of the first birth. An important situation for prevention is the surgical removal of the ovaries, aiding in the protection against breast cancer significantly [25].

People with the diagnosis of cancer find various forms of treatments, surgery, radiotherapy and therapies available [26]. The prognosis for breast cancer has a very high diversification according to the function, extent of disease and age of the patient. Historically, white women had the highest rates of disease incidence among women aged 40 years or older. Incidence rates have increased for estrogen receptor positive breast cancer in younger white women, Hispanic women aged 60–69, and all but the older African-American women [27].

Breast cancer is the type of cancer that affects women more; however, there is a small possibility of occurring in men, even in a very unusual way, since according to statistics, for every 1 man diagnosed with cancer 100 women present the disease. In the United States, 1500 new cases are identified each year [28]. Figure 1 highlights the characteristics of cancer in a mammary region.

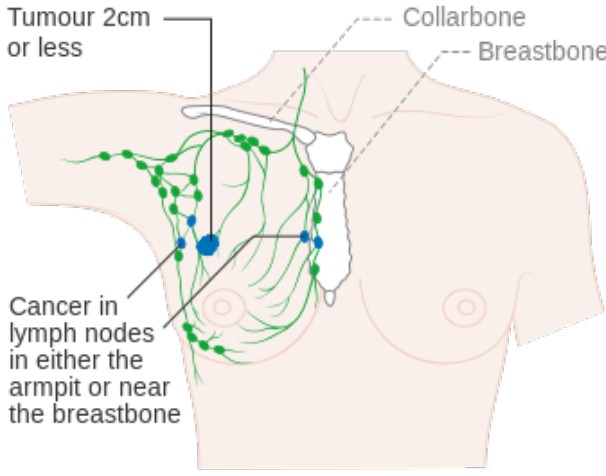

**Figure 1.** Breast cancer [29].

## 2.2. Artificial Neural Networks

Artificial neural networks are models that apply the logical neuron to their structures, thus simulating the processing of information from the human brain through a network of innumerable artificial neurons, which interconnect through synaptic connections. Briefly, an artificial neural network can be compared to a graph, where the nodes are the neurons and the links function the synapses. The way in which the weights are adjusted during the learning process greatly differentiates the architecture of the neural networks. One of the main factors is learning, where the artificial neural network captures the information supplied by the inputs, through the connections and the synaptic weights. The neural network is determined by the number of layers being single or multiple layers. It is also defined by the number of nodes in each layer, the type of connection between nodes (feedforward or feedback) and their number of responses [30].

Figure 2 below shows a neural network structure with multiple layers. It also shows the neurons and their synaptic connections.

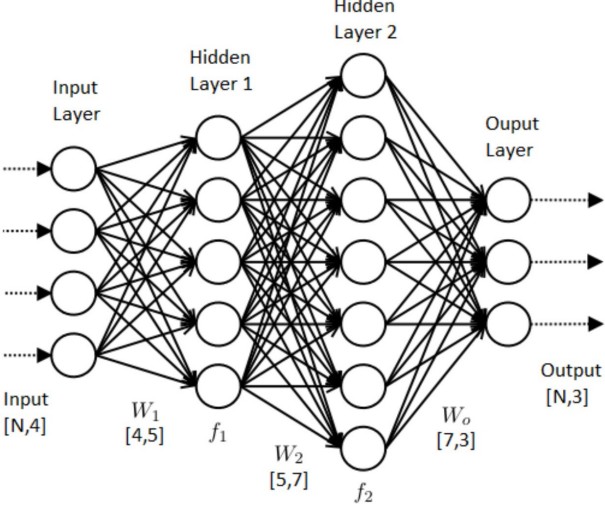

**Figure 2.** Artificial neural network of multiple layers and outputs [31].

*2.3. Fuzzy Systems*

The use of the fuzzy system becomes essential in situations where the classical approach becomes infeasible for the solution of a problem as a consequence of the essence of its complexity. Fuzzy systems have features: membership functions, rules, and aggregation operators, which provides a closer approximation to the actual model. Due to this approximation, it is avoided that resolution of the fuzzy system differs significantly from the expected reality [32]. The fuzzy system has the ability to identify, reproduce, manipulate, interpret, and apply information and data that is vague and uncertain. Figure 3 presents the main elements that integrate fuzzy logic: Inputs, the process of modifying inputs in fuzzy elements, the formation of fuzzy input sets, the grouping of rules and inferences, obtaining fuzzy response sets, which is to convert a numerical value into a fuzzy set using the membership functions [8].

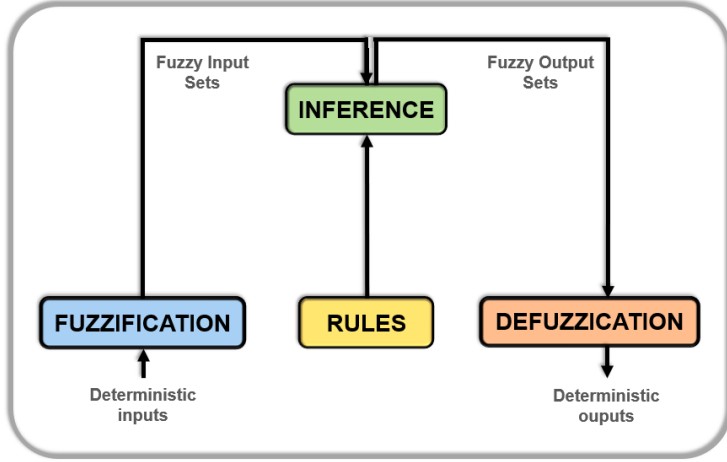

**Figure 3.** Concepts present in fuzzy logic [33].

## 2.4. Fuzzy Logic Neurons

Many studies have been done to emulate the behavior of human neurons, evidencing those that sought to add the fuzzy nature to the artificial neuron model, complementing the ability to treat imprecise information. This neuron we call a fuzzy neuron [34]. Type III fuzzy neurons (represented by fuzzy logic equations) provide some variations of models that combine fuzzy type II neurons (with fuzzy inputs combined with fuzzy weights) and III. To these neurons we call logical neurons. They were proposed by [35] and have a structure similar to type III neurons, except that the connection weights $w_i$ are associated with each of the inputs $x_i$. Thus the logical neuron performs a mapping in the space formed by the Cartesian product between the input space and the space of the weights in the unit interval, that is **X** x **W** − [0,1] [34].

We can point to logical fuzzy *and* and *or* logical neurons, as well as a one-way-based generalization of t-norm and s-norm [36] , calling unineuron. When we use the logical neuron and we use a s-norm as a weighting operator, while the aggregation operator is implemented by a t-norm, unlike when we use the which is accomplished through the use of a t-norm as a weighting operator and a norm as an aggregation operator [34]. Once unineuron was initially treated as the extension of logical and neurons and where the neuron processing takes place on two levels. In the first level the input signals are individually combined with the weights, already in the second level a global action is performed on the results of all combinations of the first level [37]. Consider as the input signal $[a_1, a_2, ...a_n]$ and the weights $[w_1, w_2, ...w_n]$ for $a_i \in [0, 1]$ and $w_i \in [0,1]$ for $i$ of 1, ..., $n$. The aggregation performed by the fuzzy logic neurons and, Where the input signals are combined individually with the weights and carried out the subsequent global aggregation can be defined as follows [35]:

$$\mathbf{z} = OR(a, w) = S_{i=1}^n (a_i t w_i) \tag{1}$$

$$\mathbf{z} = AND(a, w) = T_{i=1}^n (a_i s w_i) \tag{2}$$

where $T$ and $t$ are the representation of a t-norm and $S$ and $s$ an s-norm. Now for unineuron [37] described the steps to perform the functions of the neuron:

- Transform each pair $(a_i, w_i)$ into a single value $b_i = h (a_i, w_i)$;
- Calculate the unified aggregation of the transformed values **U** $(b_1, b_2, ..., b_n)$, where $n$ is the number of input.

The function $h$ is responsible for transforming the inputs and corresponding weights into individual transformed values [37]. A formulation for the $h$ function can be visualized:

$$h(w, a) = w \times a + w \times g \tag{3}$$

Using the weighted aggregation reported in [37] we can write unineuron as:

$$\mathbf{z} = UNI(w, x; g) = U_{i=1}^n h(w, x) \tag{4}$$

## 2.5. Related Work

Artificial intelligence has already presented several works to solve problems of prediction of breast cancer. These works involve different resolutions through the identification of images [38–41], artificial neural networks [42–47], fuzzy approach [48,49], support vector machines [50,51], deep learning [52,53] among others.

The work proposed by [7] has already been worked on some models recently. The models of [54] which also uses a fuzzy neural network to detect breast cancer patterns, but unlike the approach proposed in this paper, a regularized approach based on resampling is used. In the work done by [55] the database

is evaluated through numerous machine learning techniques, as well as comparisons with other databases with data related to this disease. In the work of [56] the database first went through a normalization process based on the MAD technique. In the second step, resource weighting based on k-means clustering (KMC) was used to weight the normalized data. Finally, the AdaBoostM1 classifier was used to classify the weighted data set. Finally, a PhD thesis [57] addressed the problem solving and a recent paper proposed by [58] seeks improvements in the classification of artificial neural network models through new training algorithms.

## 3. Fuzzy Neural Network for Detection of Breast Cancer

*Neural Network Neural Network Architecture and Training for Binary Classification Problems*

In the field of computational intelligence the great advance in the development of techniques, models that simulate human behavior acting in systems and processes, with emphasis on artificial neural networks, fuzzy systems and their hybrid models, with an enormous amount of new applications being proposed in the literature [59]. One of the purposes of computational intelligence research is the ability to create and model computational systems that mimic peculiar characteristics of human beings, such as learning, logical reasoning, intuition, regression, and classification [30].

The fuzzy neural networks, is a mathematical model used to perform simulations of human behavior having in its structure fuzzy neurons. This model differs from the traditional models of neural networks and fuzzy systems because of the co-operation between the neural networks concepts and the fuzzy associations ideas, resulting in a model with the learning capacity of the neural networks together with the efficiency in the treatment the imprecision and the interpretation that exists in fuzzy systems. [35].

The architecture of the fuzzy neural network used in this work is the same as reported in [20]. Therefore the fuzzy neural network to be used in this paper uses in the first layer the division of input space in a grid format called FIS [60], that is, it does the fuzzification process reported in Figure 3 using the space division of input data into a number *M* of fuzzy membership functions of the type Gaussian and Triangular equally sparse and centered in 0.5. These types of divisions through membership functions with the same characteristics can aid in semantic interpretation. The fuzzy Inference System (FIS) is a computational framework based on the concepts of fuzzy set theory, fuzzy rules of the style *IF... THEN* and fuzzy reasoning where its structure has three conceptual layers: a rule base, a database, and a reasoning basis [34]. In [34] explains that the FIS can perform a non-linear mapping from its input space to the output space. This mapping is accompanied by many fuzzy IF... THEN rules, where each one describes the local behavior of the mapping. The antecedents in the rules of a fuzzy inference system implement a multidimensional neural partition, which can be in the grid, decision tree or by grouping, in the space of the input variables of a model. In this paper, an evaluation was used to filter the neurons that will leave the first layer: If the number of characteristics of the problem is less than six and the number of pertinence functions chosen are less than or equal to three, the initially proposed in [34] This choice is due to several previous tests that verified that the ability to create fuzzy rules made the problem much more complicated than was necessary. If the premise is not met, the amount of neurons in the first layer is defined using a modified proposal of [34], where there is a random-creation limiting filter of 500 fuzzy neurons based on the input space. The number 500 was arbitrated after several algorithm performance tests using cross-validation concepts with various combinations of feature numbers and membership functions. Figure 4 below explains how partitioning the input data works according to the partition used.

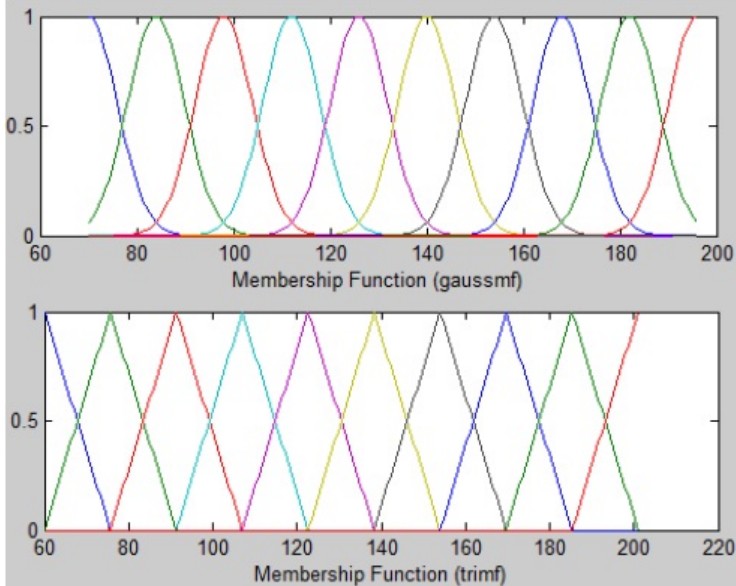

**Figure 4.** Division of input space performed by Gaussian and triangular membership functions.

Considering *v1* and *v2* as two characteristics present in the data collected from study patients, we can infer linguistic rules of the type: If *v1* is small and *v2* is small, **y** is 1, just as if *v1* and *v2* were large, the class of **y** is 2. Within this fuzzy rule model, *v1* and *v2* by elements of a cancer research base and responses 1 and two by people with cancer and people without cancer. In this context one can apply, for example, the value of *v1* a dimension as age and *v2* as the BMI of women. However, as described in [7], the database groupings were made from rules of statistical membership, and the final results of the classifier tests did not calculate the test accuracy and the interpretability of the results, highlighting only the value AUC, specificity and sensibility.

In the second layer there are fuzzy logical neurons that use unineuron concepts proposed in [37], which perform the aggregation of the weights and activation functions of the fuzzy neurons of the first layer. Fuzzy rules can be extracted from unineuron. See more in [37].

As the Anfis model can generate a high amount of rules and some of them are unnecessary for the context, the [20] approach uses in the second layer the pruning of neurons less significant for the background evaluated using the F-scores technique developed by [61]. It is capable of assessing the discriminating power of the variables of a set of characteristics. In this context, we used the concept of F-scores in the second layer of fuzzy neural network seeking to discard neurons that have smaller F-score than a defined threshold value (defined like [20]).

In this context, the calculation of the F-score will be used in the columns of **z**. To determine which elements will be selected within this approach, the criterion will be adopted in [62] and replicated in the where the $s_j$ variables are smaller than theaverage of all the f-scores found. This approach willallow a choice of the most significant neurons (candidate neurons) ($L_s$) participating in the fuzzy neural network response [20].

The second layer weights are defined using the extreme learning machine concepts [23] that act efficiently in fast weight generation, unlike methods that work with backpropagation, method [63] to update the network architecture. In models that work with large data mass, this approach becomes feasible due to its more straightforward nature of generating the weights using pseudo-inverse [64] concepts. This

approach is used in several contexts in science to solve problems related to the pattern classification [65–67], time series forecasting [68–70], industry issues [71–74], health area [75–77], among others.

The learning algorithm assumes that the output hidden layer composed of the candidate neurons can be written as [20]:

$$f(x_i) = \sum_{i=0}^{L_s} v_i z_i(x_i) = sign(f(\mathbf{z}(x_i)\mathbf{v}))$$ (5)

where $\mathbf{v} = [v_0, v_1, v_2, ..., v_{Ls}]$ is the weight vector of the output layer and $\mathbf{z}(x_i) = [z_0, z_1(x_i), z_2(x_i)...z_{Ls}(x_i)]$ the output vector of the second layer, for $z_0 = 1$ and sign is a step function that transforms values greater than zero into 1 and values smaller than zero into $-1$. In this context, $\mathbf{z}(x_i)$ is considered as the non-linear mapping of the input space for a space of fuzzy characteristics of dimension $L_s$ [20]. The learning algorithm has only to estimate the output layer vector $\mathbf{v}=[v_0, v_1, v_2...v_L]^T$ which best adjustment the wanted output. In this paper these parameters are computed using the Moore-Penrose pseudo-inverse [64]:

$$\mathbf{v} = \mathbf{Z}^+\mathbf{y}$$ (6)

$Z^+$ is the Moore-Penrose pseudo Inverse [64] of $\mathbf{z}$, which is the minimum norm of the least squares solution for the output weights.

Finally, the model uses a simple artificial neuron in the third layer, which can be seen as singleton. The architecture of the fuzzy neural network used in the paper is presented in Figure 5. The following Algorithm 1 presents the necessary steps for the execution of the classification of patterns on the treatments for breast cancer. It has two parameter:

1. the number of membership functions, $M$;
2. the type of fuzzy logic neuron, unineuron, andneuron or orneuron;

---
**Algorithm 1:** FNN- Training

---
(1) Define the number os membership functions, $M$.
(2) Calculate $M$ neurons for each characteristic in the first layer using Anfis.
(3) Construct $L$ fuzzy neurons with Gaussian or Triangular membership functions constructed with center and $\sigma$ values derived from ANFIS (Using genfis1 approach).
(4) Define the weights and bias of the fuzzy neurons randomly.
(5) Construct $L$ fuzzy logical neurons with random weights and bias on the second layer of the network by welding the $L$ fuzzy neurons of the first layer.
(6) Use f-scores to define the most significant neurons to the problem ($L_s$).
(7) **For all** $K$ input do
(7.1) Calculate the mapping $z_k(x_k)$ using logical neurons
**end for**
(8) Estimate the weights of the output layer using Equation (6).
(9) Calculate output $\mathbf{y}$ using Equation (5).

---

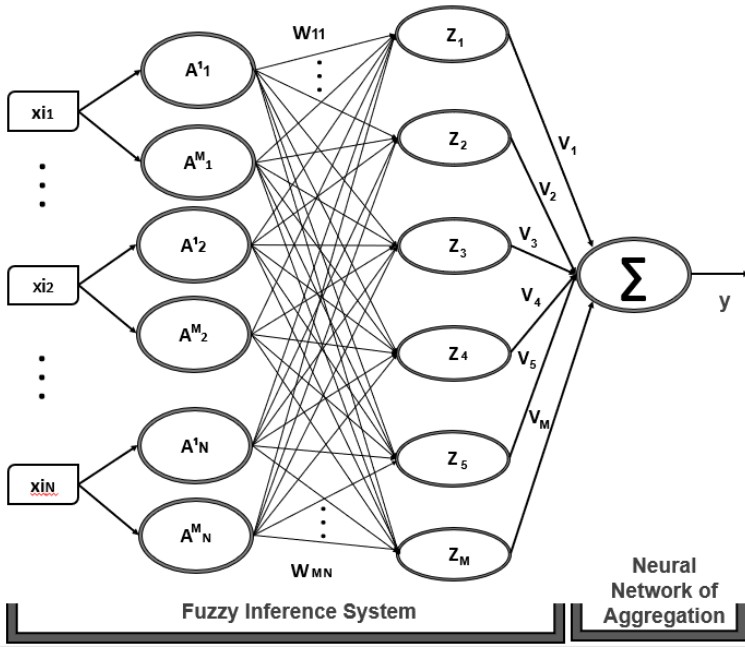

**Figure 5.** Structure of the fuzzy neural network used in the paper [78].

## 4. Patient Classification Tests Using Fuzzy Neural Network

### 4.1. Dabase on Breast Cancer Prediction Research

The research for the prediction of breast cancer (The database used in the tests can be obtained from https://archive.ics.uci.edu/ml/datasets/Breast+Cancer+Coimbra) presented in [7] was performed after a study in which 9 attributes were used as parameters, in the participation of 166 patients. These parameters were organized and submitted to three classification algorithms: logistic regression, random forests and support vector machines. The values collected in the study were organized using letters and numbers to represent characteristics, where:

$V1$ = Glucose
$V2$ = Resistin
$V3$ = Age
$V4$ = BMI − body mass
$V5$ = HOMA − evaluation of the homeostasis model for insulin resistance
$V6$ = Leptin
$V7$ = Insulin
$V8$ = Adiponectin
$V9$ = MCP-1 − monocyte-1 chemotactic protein.

For modeling, the author divided the variables into 6 clusters according to statistical membership techniques (V1-V2, V1-V3, V1-V4, V1-V5, V1-V6, V1-V9). Each combination considered as the predictor characteristics the most significant dimensions for the problem, calculated from a feature selection rule to be able to determine what were the most relevant factors to determine patients with cancer probability. In correlation with the classification, there are three evaluation algorithms: AUC, Sensitivity and Specificity. Where sensitivity and specificity consist respectively in guessing when sick people are really sick and not sick when they do not have the disease. Based on the results, the presence or absence of breast cancer in

women could be predicted with sensitivity ranging from 82% to 88% and specificity ranging from 85% to 90%. Indicating that Resistin and Glucose, together with Age and BMI, can be considered a good grouping for biomarkers of breast cancer.

The worst AUC intervals were obtained in the Random Forest model in the V1-V2 combination ([0.70, 0.75]), sensitivity occurred in the Logistic Regression model in the combination V1-V9 ([0.70, 0.76]). Finally, the specificity was found to be lower concerning Random Forest V1-V2 ([0.63, 0.70]).

The best AUC results were found in the Support Vector Machine and V1-V4 test([0.87, 0.91]). For Sensitivity, the best value interval was the SVM model in the V1-V3 analysis ([0.87, 0.92]). Finally, Specificity achieved the best results also with the SVM but in the V1-V4 test ([0.84, 0.90]).

### 4.2. Test Configuration

The tests performed in the database were used in the regularized and non-regularized versions of the [60] model. The pertinence functions are of the Gaussian and Triangular types, the first and second layer weights were randomly estimated in the range of $[-3, 3]$.The neurons used in the tests were unineuron and andneuron. The model resolution is calculated by checking the network output with the expected classification value. The sum of these values is divided by the total of samples used in the training and test steps to determine the accuracy, respectively. More detailed information on [60] and [37]. To avoid tendency, 30 repetitions with random inputs were performed. The training target is 70 % for training and 30 % for testing. The number of membership functions of the fuzzy neural network models were defined using the cross-validation procedure with k-fold technique. For this purpose, 70 % of the training samples and 30 % of the samples were used for validation with all combinations of M = [3, 4, 5, 6, 7]. The result of the cross-validation test evaluated the best value of accuracy, selecting the best value of M. The outputs of the model were normalized to $-1$ and 1 to aid the correct calculations. The factors evaluated in this paper are as follows:

$$accuracy = \frac{TP + TN}{TP + FN + TN + FP} \tag{7}$$

$$sensitivity = \frac{TP}{TP + FN} \tag{8}$$

$$specificity = \frac{TN}{TN + TP} \tag{9}$$

$$AUC = \frac{1}{2}(sensitivity + specificity) \tag{10}$$

where, $TP$ = true positive, $TN$ = true negative, $FN$ = false negative and $FP$ = false positive.

### 4.3. Patient Classification Tests

The pattern classification test was performed with breast cancer patients and presented two test cores. Figure 6 shows the neural network performance test using the unineuron with Gaussian membership functions (Uninet), the andneuron with triangular membership functions (Andnet) and orneuron and Gaussian membership functions (OrNet). The values of the membership function are in the range of [2–5] (defined by cross-validation with 70% training and 30% validation with 10 k-fold). On the other hand, the values of the training accuracy and test accuracy are presented in average percentage values after the execution of the 30 repetitions. The values shown in parentheses are the standard deviations for the experiment. It is also emphasized the values of Sensitivity, Specificity and AUC, in addition to the amount

resulting from neurons used after regularization. In all the tables presented in the tests the collected time is expressed in seconds. Figure 6 shows the values obtained in the breast cancer classification tests.

| Data-sets | Model | amount of neurons pruned | Sensitivity | Specificity | AUC | Accuracy | test time |
|-----------|-------|--------------------------|-------------|-------------|-----|----------|-----------|
| V1V2 | AndNet | 24,83 (11,96) | 74,13 (12,11) | 70,27 (9,69) | 0,72 (0,07) | 72,1 (7,45) | 0,5 (0,52) |
| | OrNet | 26,47 (11,55) | 73,79 (11,29) | 68,68 (16,49) | 0,72 (0,06) | 71,81 (5,9) | 0,55 (0,51) |
| | UniNet | 25,7 (11,78) | 72,8 (17,22) | 66,59 (20,59) | 0,7 (0,05) | 70,29 (6,21) | 0,51 (0,51) |
| V1V3 | AndNet | 86,93 (6,8) | 81,45 (7,88) | 72,45 (9,06) | 0,76 (0,06) | 77,33 (5,95) | 1,76 (0,28) |
| | OrNet | 84,7 (4,71) | 78 (10,48) | 71,71 (16,82) | 0,75 (0,07) | 74,95 (6,94) | 1,94 (0,25) |
| | UniNet | 83,63 (5,94) | 78,68 (10,49) | 69,37 (15,7) | 0,74 (0,07) | 74,48 (6,83) | 1,81 (0,27) |
| V1V4 | AndNet | 86,37 (5,67) | 83,28 (8,5) | 78,25 (11,94) | 0,8 (0,07) | 80,48 (6,41) | 1,75 (0,28) |
| | OrNet | 87,4 (5,35) | 79,92 (10,14) | 81,56 (10,54) | 0,8 (0,07) | 80,67 (6,03) | 1,96 (0,24) |
| | UniNet | 86,3 (6,33) | 81,44 (10,01) | 70,66 (17,34) | 0,76 (0,07) | 76,95 (7,08) | 1,84 (0,27) |
| V1V5 | AndNet | 55,57 (7,04) | 76,53 (10,04) | 74,89 (11,22) | 0,75 (0,06) | 75,52 (5,75) | 2,2 (0,22) |
| | OrNet | 53,43 (7,77) | 71,67 (9,12) | 72,95 (11,43) | 0,72 (0,06) | 71,91 (5,52) | 2,32 (0,19) |
| | UniNet | 45,67 (4,74) | 82,46 (24,29) | 24,1 (31,34) | 0,55 (0,08) | 54,57 (8,23) | 0,81 (0,16) |
| V1V6 | AndNet | 57,17 (6,47) | 75,38 (20,29) | 54,06 (32,31) | 0,65 (0,1) | 64,76 (10,24) | 2,46 (0,16) |
| | OrNet | 53,63 (7,46) | 78,2 (15,74) | 51,47 (33,41) | 0,65 (0,11) | 66,38 (10,5) | 2,83 (0,15) |
| | UniNet | 49,1 (8,49) | 75,2 (14,78) | 55,76 (27,91) | 0,67 (0,08) | 67,43 (7,45) | 2,48 (0,15) |
| V1V7 | AndNet | 56,53 (6,51) | 70,92 (23,85) | 61,58 (26,51) | 0,67 (0,08) | 66,19 (8,39) | 2,96 (0,11) |
| | OrNet | 56,13 (6,77) | 75,23 (12,42) | 65,96 (21,61) | 0,71 (0,07) | 71,14 (6,89) | 3,21 (0,1) |
| | UniNet | 52,1 (7,28) | 78,91 (11,36) | 64,19 (23,51) | 0,72 (0,07) | 73,43 (6,42) | 3,02 (0,1) |
| V1V8 | AndNet | 43,43 (4,49) | 85,11 (19,46) | 21,82 (28,6) | 0,57 (0,07) | 56,38 (7,19) | 1,04 (0,08) |
| | OrNet | 55,63 (5,66) | 71,31 (30,23) | 38,77 (35,74) | 0,56 (0,11) | 56,04 (10,8) | 1,2 (0,09) |
| | UniNet | 43,73 (3,98) | 83,42 (23,32) | 24,38 (28,73) | 0,56 (0,08) | 56,29 (8,43) | 1,07 (0,1) |
| V1V9 | AndNet | 58,93 (4,68) | 76,46 (40,25) | 24,51 (39,61) | 0,51 (0,08) | 53,05 (6,08) | 0,67 (0,05) |
| | OrNet | 54,3 (5,08) | 76,01 (12,25) | 45,93 (18,96) | 0,63 (0,06) | 62,86 (6,88) | 0,82 (0,03) |
| | UniNet | 46,73 (4,18) | 78,36 (19,77) | 41,24 (25,25) | 0,62 (0,07) | 62 (7,91) | 0,7 (0,04) |

**Figure 6.** Test Performance Fuzzy Neural Network.

In the figure shown, one can observe an evident approach on the concepts of AUC and accuracy. The model supported the approximate AUC to the experiments proposed in [7], contrasting by using an average of 30 replicates. The specificity and sensitivity of the model also helps in the understanding of how the neural network can work with the construction of fuzzy rules avoiding false negative positives. In the first experiment, 3 membership functions were used, each assigning a literal value, such as "small", "medium" and "large". Based on the membership functions, it is possible to compare the concepts of Glucose and Resistin by means of IF/THEN rules, thus identifying the literal level of each of them according to the patient, determining a correlation that can be used for the elaboration of expert systems.

After the preliminary tests, we identified the database that the FNN model obtained the best results (V1-V4) and chose this base to undergo a comparative analysis with five intelligent models that are available in the WEKA tool [79]. Among the models chosen were Naive Bayes (NB) [80], the Multilayer Perceptron (MLP) [81], Random Tree [82], Zero R [83], and a decision tree model called C 4.5 [84]. These models were chosen because they are commonly used as standard classifiers and are highlighted in several scientific articles.

The values used in the fuzzy neural network models were the same as in the previous test. For WEKA [79], the experiment tool was used, allocating 70% of the samples for training and 30% for tests. The parameters follow the initial configuration proposed by the device. The procedure of 10 k-fold was used in the tool to obtain the results. Table 1 shows the values obtained in the breast cancer classification tests.

**Table 1.** Results V1-V4 database.

| Models | Accuracy | AUC | Sensitivity | Specificity | Time |
|--------|----------|-----|-------------|-------------|------|
| **AndNet** | 80.01 (6.98) | **0.8052 (0.1011)** | 0.7841 (0.1124) | 0.7905 (0.1003) | 1.94 (0.03) |
| **OrNet** | **81.04 (4.85)** | 0.8019 (0.0918) | 0.8193 (0.1024) | **0.8118 (0.1203)** | 1.89 (0.02) |
| **UniNet** | 78.49 (4.97) | 0.7624 (0.1207) | **0.8248 (0.1412)** | 0.7105 (0.0311) | 2.19 (0.06) |
| **MLP** | 73.80 (11.38) | 0.7409 (0.1206) | 0.4187 (0.2102) | 0.7803 (0.1018) | 15.62 (6.04) |
| **J48** | 71.83 (14.27) | 0.7114 (0.1243) | 0.5276 (0.1206) | 0.7895 (0.1032) | **0.78 (0.01)** |
| **NB** | 69.71 (12.61) | 0.6925 (0.1213) | 0.3687 (0.1786) | 0.7112 (0.2142) | 15.22 (0.76) |
| **ZR** | 55.15 (3.10) | 0.5004 (0.0012) | 0.5500 (0.0021) | 0.5500 (0.0016) | 7.12 (0.43) |
| **RT** | 79.67 (11.67) | 0.7421 (0.1120) | 0.5301 (0.1703) | 0.8718 (0.0613) | 8.21 (0.01) |

We can see that after the execution of the tests the models with the fuzzy neural network pruned were the ones that presented the best general results. The results with the traditional literature models also corroborate that the database is very complex and it becomes increasingly complicated to work with it. In this case, intelligent models can extract knowledge from the network to further assist in feature selection techniques and other approaches to improve the quality of algorithm evaluation.

## 5. Conclusions

After the experiments, one can arrive at some significant results. There was a considerable increase in accuracy, sensitivity, and specificity concerning the work was done by [7]. In addition to a substantial improvement in results, the model is calculated in a scientific way and without trend. Another essential point to be mentioned is that about the work done by [54], the pruning method has much faster performance than the resampling method because it does everything in one step. This made the model more compact and with very low execution time.

The works based on fuzzy neural networks work with approaches that can lead to the creation of a set of fuzzy rules to build expert systems to serve as tools for physicians who deal with this complex subject.

In future works, other fuzzy neural models may be used, and more efficient algorithms for the selection of characteristics for the treatment of breast cancer, as well as other pathologies.

**Author Contributions:** Conceptualization, V.J.S.A. and P.V.d.C.S.; methodology, A.J.G.; software execution, P.V.d.C.S. and T.S.R.; validation, V.S.A., V.J.S.A.; formal analysis, P.V.d.C.S.; investigation, A.J.G. and V.J.S.A.; data curation, P.V.d.C.S.; writing–original draft preparation, V.J.S.A. and P.V.d.C.S. and A.J.G.; writing–review and editing, V.J.S.A., A.J.G. and P.V.d.C.S.; visualization, V.S.A. and T.S.R.; supervision, P.V.d.C.S.; project administration, V.J.S.A. and P.V.d.C.S.

**Funding:** This research received no external funding.

**Acknowledgments:** Acknowledgments to this work are destined to the UNA University Center and the Federal Center of Technological Education of Minas Gerais-CEFET-MG.

**Conflicts of Interest:** The authors declare no conflict of interest.

## Abbreviations

The following abbreviations are used in this manuscript:

FNN    Fuzzy Neural Network
AUC    Area Under Curve
UNI    unineuron
AND    andneuron

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
