# Peer review of "Using Resistin, Glucose, Age and BMI and Pruning Fuzzy Neural Network for the Construction of Expert Systems in the Prediction of Breast Cancer"

_make, doi:10.3390/make1010028_

Round 1

Reviewer 1 Report

Thank you for improving the manuscript

Author Response

Dear Reviewer. Thank you very much for the compliments and the valuable contributions to our work.

Kind regards.

Paulo Vitor

Reviewer 2 Report

They have improved the paper. However, they need much more work tobe addressed.

 1- There is an extra section included in the paper with no content. I suggest removing it and renumber the others.

"4. Testing Identification of patients able to treatment cryotherapy and immunotherapy".

2- There are two tables 1 in the paper.Furthermore, thecaptionsare locateddifferently for each other (top and bottom). Table's call is missing in the manuscript.

3- The last paragraph of introduction is missing Section 4. This correct, but in discordance of the current version which presents section 4.

4- There is no accordance between reference [54]. Inside the text, it should be master's thesis whereas in the Bio is PhD's thesis.

5- I suggest a careful revision of references, figures, tables, sections, citation, etc. The template presents problems.

6- There are more tobe done, the above comment was examples.

Author Response

Dear reviewer, how are you?

Many thanks for the contributions. I believe that some misunderstandings have occurred because of the lack of familiarity with the latex, but we are providing the improvements.

1- There is an extra section included in the paper with no content. I suggest removing it and renumber the others.

"4. Testing Identification of patients able to treatment cryotherapy and immunotherapy".

Thank you for the tip. This label went unnoticed. The extra section has been deleted and the numbering properly corrected.

2- There are two tables 1 in the paper.Furthermore, thecaptionsare locateddifferently for each other (top and bottom). Table's call is missing in the manuscript.

The table numbering and reference have been updated.

3- The last paragraph of introduction is missing Section 4. This correct, but in discordance of the current version which presents section 4.

The numbering has been corrected as soon as we exclude the excess section. Thanks for the tip.

4- There is no accordance between reference [54]. Inside the text, it should be master's thesis whereas in the Bio is PhD's thesis.

The term was duly corrected in the text

5- I suggest a careful revision of references, figures, tables, sections, citation, etc. The template presents problems.

6- There are more to be done, the above comment was examples.

To meet the request, we evaluate all errors and warnings presented by latex. All have been fixed.

I believe the document is now ready for publication. Many thanks for the excellent review.

Round 2

Reviewer 2 Report

The authors have improved the paper properly. There is only one point to be addressed. The table's captions are different. One is on the bottom and another is on the top. However, the authors could correct it during the proof reading procedure.

This manuscript is a resubmission of an earlier submission. The following is a list of the peer review reports and author responses from that submission.

Round 1

Reviewer 1 Report

I believe this to be an interesting article on an important topic. I believe the novelty of the article lies in the fact that machine learning techniques have still not been widely adopted to handle the very relevant problem of breast cancer or similar problems. In my opinion, understanding what techniques fit the problem the best is an important contribution to Science and to Medicine.

1- My major concern about the article is the claim in the conclusion "There was a considerable increase in accuracy, sensitivity, and specificity concerning the work was done by [7]." This is followed, in the same conclusion, by "After the experiments we arrived at the conclusion that even reaching a lower value of the accuracy in relation to the work of [7], the results have their relevance for science, because the accuracy is calculated in a scientific and without tendencies.

The conclusion is thus very unclear:

a) either there is a considerable increase in accuracy or there is a decrease in accuracy

b) When you say "the accuracy is calculated in a scientific and without tendencies", does it mean that you found limitations in the methodology of [7]? If so, I believe you should explicitly say which limitations you were able to avoid by your approach. 

2-If I understand correctly, the main advantage of this article when compared to [7] is twofold: the interpretability of the model and good prediction accuracy. I can understand this clearly when reading the article but not when reading the abstract. I believe the abstract should be rewritten in a more clear way.

3- I believe section 2.1 can be removed as it is not relevant for the article

4- On Figure 6 it is not possible to see which classifier algorithms the results correspond to. In any case, I would not include the figure - instead I would mention the sensitivity, specificity and AUC confidence intervals of the worst models and the best models.

5- Your formula (11) expresses the AUC as the average between the sensitivity and specificity. However, AUC is the area under the ROC curve - how is this transformed into this average? Is this an approximation?

Reviewer 2 Report

The paper presents a hybrid model to manipulate the data collected in medical examinations and identify patterns among healthy people and people with breast cancer.The method is basedon Neural Network and Fuzzy. The paper presentflaws as follows:

Minor Comments:

1- At line 62, it sounds that is missing a reference citation there;

2- At lines 132 and 127,the references are foundafter the period;

3- References 50 and 55 seem to be incomplete;

4- At page 178, "figure" instead of "Figure";

5- The testing time is in seconds? Milliseconds?

6- Figure 6 should be namedas a table;

Major Comments:

1 -  I miss a comparison analysis for the proposed method to others found in the literature. They mentioned reference [7], but where is the comparison? I suggest the authors insert a comparison analysis in the manuscript. How could weevaluatethe advances achieved in their methods without metrics and time comparison?

2- The whole procedure is confusing because of several paragraphs that try to explain the method. How is a sequence of the method? To be honest, I could see the method.In my opinion, theauthors only presented "Theoretical aspects" instead. I suggest a block diagram for the proposed method;

3- The results have presented a low accuracy rate, mostly below 75%.